  

# Real-time detection of condensin-driven DNA compaction reveals a multistep binding mechanism

Jorine M Eeftens[1], Shveta Bisht[2], Jacob Kerssemakers[1], Marc Kschonsak[2], Christian H Haering[2,*] & Cees Dekker[1,**]

## Abstract

Condensin, a conserved member of the SMC protein family of ring-shaped multi-subunit protein complexes, is essential for structuring and compacting chromosomes. Despite its key role, its molecular mechanism has remained largely unknown. Here, we employ single-molecule magnetic tweezers to measure, in real time, the compaction of individual DNA molecules by the budding yeast condensin complex. We show that compaction can proceed in large steps, driving DNA molecules into a fully condensed state against forces of up to 2 pN. Compaction can be reversed by applying high forces or adding buffer of high ionic strength. While condensin can stably bind DNA in the absence of ATP, ATP hydrolysis by the SMC subunits is required for rendering the association salt insensitive and for the subsequent compaction process. Our results indicate that the condensin reaction cycle involves two distinct steps, where condensin first binds DNA through electrostatic interactions before using ATP hydrolysis to encircle the DNA topologically within its ring structure, which initiates DNA compaction. The finding that both binding modes are essential for its DNA compaction activity has important implications for understanding the mechanism of chromosome compaction.

**Keywords** condensin; DNA compaction; magnetic tweezers; SMC proteins
**Subject Categories** Cell Cycle; Chromatin, Epigenetics, Genomics & Functional Genomics; Structural Biology
The EMBO Journal (2017) 36: 3448–3457

## Introduction

The structural maintenance of chromosome (SMC) complexes cohesin and condensin play central roles in many aspects of chromosome biology, including the successful segregation of mitotic chromosomes, chromatin compaction and regulation of gene expression (reviewed in Hirano, 2016; Aragon *et al*, 2013; Nasmyth & Haering, 2009). SMC protein complexes are characterized by their unique ring-like structure (Fig 1A). The architecture of condensin is formed by a heterodimer of Smc2 and Smc4 subunits, which each fold back onto themselves to form ~45-nm-long flexible coiled coils, with an ATPase "head" domain at one end and a globular "hinge" heterodimerization domain at the other end (Anderson *et al*, 2002; Eeftens *et al*, 2016). The role of ATP binding and hydrolysis by the head domains has remained largely unclear. The head domains of the Smc2 and Smc4 subunits are connected by a protein of the kleisin family, completing the ring-like structure (Fig 1A). The condensin kleisin subunit furthermore recruits two additional subunits that consist mainly of HEAT-repeat motifs. Most metazoan cells express two condensin complexes, condensin I and II, which contain different non-SMC subunits and make distinct contributions to the formation of mitotic chromosomes (Ono *et al*, 2003). The genome of the budding yeast *Saccharomyces cerevisiae,* however, encodes only a single condensin complex, which contains the kleisin subunit Brn1 and the HEAT-repeat subunits Ycg1 and Ycs4 (Fig 1A).

How condensin complexes associate with chromosomes has remained incompletely understood. Biochemical experiments have provided evidence that condensin, similar to cohesin, embraces DNA topologically within the ring formed by the Smc2, Smc4 and Brn1 subunits (Cuylen *et al*, 2011). In addition, the HEAT-repeat subunits were found to contribute to condensin's loading onto chromosomes and the formation of properly structured chromosomes (Piazza *et al*, 2014; Kinoshita *et al*, 2015). In contrast, ATP hydrolysis by the Smc2–Smc4 ATPase heads does not seem to be absolutely required for the association of condensin with chromosomes *in vivo* (Hudson *et al*, 2008) or in extracts (Kinoshita *et al*, 2015) and condensin binds DNA *in vitro* even in the absence of ATP (Kimura & Hirano, 1997; Strick *et al*, 2004). DNA, can, however, stimulate ATP hydrolysis by the Smc2–Smc4 ATPase heads (Kimura & Hirano, 2000; Piazza *et al*, 2014). These findings have led to the speculation that condensin might initially bind to the DNA double helix by a direct interaction, possibly with its HEAT-repeat and kleisin subunits, and that this binding might subsequently trigger an ATP hydrolysis-dependent transport of DNA into the condensin ring (Strick *et al*, 2004; Piazza *et al*, 2014; Kschonsak *et al*, 2017). This hypothesis has not yet been confirmed, however. The

1 Department of Bionanoscience, Kavli Institute of Nanoscience Delft, Delft University of Technology, Delft, The Netherlands
2 Cell Biology and Biophysics Unit, Structural and Computational Biology Unit, European Molecular Biology Laboratory (EMBL), Heidelberg, Germany
 *Corresponding author. Tel: +49 6221 3878450; E-mail: christian.haering@embl.de
 **Corresponding author. Tel: +31 15 2786094; E-mail: c.dekker@tudelft.nl
 [The copyright line of this article was changed on 1 December 2017 after original online publication.]

condensin–DNA interaction is presumably the key to the mechanism by which condensin drives DNA compaction, a subject of keen interest and intense debate (reviewed in Swedlow *et al*, 2003; Thadani *et al*, 2012; Kschonsak & Haering, 2015). Models for the condensin-driven compaction of DNA include random cross-linking, condensin multimerization, and/or DNA loop extrusion (Alipour & Marko, 2012; Wilhelm *et al*, 2015; Fudenberg *et al*, 2016; Goloborodko *et al*, 2016a,b). The loop extrusion model has recently gained support, but a consensus has not yet been reached (Dolgin, 2017). Finally, condensin has also been suggested to alter the supercoiled state of DNA to promote DNA compaction (Kimura & Hirano, 1997; Kimura *et al*, 1999; Bazett-Jones *et al*, 2002; St-Pierre *et al*, 2009).

One caveat of most biochemical experiments is that they can only probe the final geometry of the DNA, but cannot address the interaction of condensin molecules with DNA during the compaction cycle. To resolve the compaction mechanism, an understanding of the binding properties of individual condensin complexes to DNA will be essential. Single-molecule techniques are especially suitable for investigating the mechanical properties, structure and molecular mechanism of SMC proteins. For example, single-molecule imaging methods proved to be crucial for revealing the sliding and motor action of individual SMC complexes on DNA (Davidson *et al*, 2016; Kim & Loparo, 2016; Stigler *et al*, 2016; Terakawa *et al*, 2017). Likewise, magnetic tweezer experiments have been successfully used to describe the compaction of DNA by the *Escherichia coli* SMC protein MukB (Cui *et al*, 2008) and by condensin I complexes immunopurified from mitotic *Xenopus laevis* egg extracts (Strick *et al*, 2004).

To obtain insights into the DNA compaction mechanism of condensin complexes, we here employ magnetic tweezers to study DNA compaction induced by the *S. cerevisiae* condensin holocomplex. Magnetic tweezers are exquisitely fit to study the end-to-end length and supercoiling state of DNA at the single-molecule level. We show real-time compaction of DNA molecules upon addition of condensin and ATP. The compaction rate depends on the applied force and the availability of protein and hydrolysable ATP. Through rigorous systematic testing of experimental conditions, we provide evidence that condensin makes a direct electrostatic interaction with DNA that is ATP independent. We further show that ATP hydrolysis is then required to render the association with DNA into a salt-resistant topological binding mode, where the DNA is fully encircled by the condensin ring. Our findings are inconsistent with a "pseudo-topological" binding mode, in which a DNA molecule is sharply bent and pushed through the condensin ring without the need to open the SMC–kleisin ring. Our results show that condensin uses its two DNA-binding modes to successfully compact DNA, thus setting clear boundary conditions that must be considered in any DNA organization model. We present a critical discussion of the implications of our results on the various models for the mechanics of condensin-mediated DNA compaction and conclude that our findings are compatible with a loop extrusion model.

# Results

## Condensin compacts DNA molecules against low physical forces

To measure the real-time compaction of individual linear DNA molecules by the *S. cerevisiae* condensin holocomplex in a magnetic

tweezer set-up, we tethered individual DNA molecules between a magnetic bead and a glass surface in a buffer condition that reflects physiological salt concentrations (Fig 1B). We then used a pair of magnets to apply force and to thereby stretch the tethered DNA molecules. We routinely performed a pre-measurement to determine the end-to-end length of the bare DNA at the force applied (Fig 1C, left of time point zero). We then simultaneously added condensin (8.6 nM) and ATP (1 mM) to the flow cell (Fig 1C, time point zero). Following a short lag time, the end-to-end length of the DNA started to decrease until, in the vast majority of cases, the bead had moved all the way to the surface. We thus observe condensin-driven DNA compaction in real time at the single-molecule level.

As different DNA tethers in the same experiment typically displayed a sizeable variation between individual compaction traces (Fig 1D), we quantitatively characterized the compaction traces using two clearly defined parameters. First, we measured the lag time, that is the time it took for compaction to initiate after adding condensin at time zero (Fig 1C). Second, starting from the decrease in the end-to-end length of the DNA, we measured the compaction rate in nanometres per second (Fig 1C). To avoid a bias at either end of the curve, we extracted the average compaction rate from the decrease between the 90 and 10% levels of the initial end-to-end length.

While keeping protein and ATP concentrations constant, we first determined compaction rates at different applied forces. We found that condensin was able to compact DNA against applied forces of up to 2 pN, albeit with rates that strongly decreased with increasing force (Fig 1E). This is surprising, since many biological motor proteins can work against forces much higher than 2 pN. On average, the rate was in the same range as measured for the *Xenopus* complex previously (Strick *et al*, 2004) and remained constant over the course of the experiment for each tether, only slowing down slightly towards the end (Fig EV1). Concurrent with the decrease in compaction rates, lag times increased with increasing force (Fig 1F). We conclude that compaction is slower and takes longer to initiate when condensin complexes are acting against a higher applied force.

The compaction rate increased approximately linearly with the concentration of the budding yeast condensin complex (Fig 1G). Higher amounts of protein were able to condense DNA much faster, at rates of up to 200 nm/s. Similarly, the lag times decreased at higher protein concentrations (Fig 1H). These findings suggest that, at higher concentrations, multiple condensin complexes might work in parallel on the same DNA molecule, resulting in faster compaction.

## DNA compaction requires DNA binding and subsequent ATP hydrolysis by condensin

We found that the compaction rate increased with increasing ATP concentrations and saturated at concentrations above a few mM (Fig 2A). A Michaelis–Menten fit resulted in a maximum compaction rate $v_{max}$ of $85 \pm 28$ nm/s (95% confidence interval) and a $K_M$ of $1.4 \pm 1.5$ mM for a protein concentration of 8.6 nM. Lag times were much longer at lower ATP concentrations (Fig 2B).

To test whether *S. cerevisiae* condensin could, like the *X. laevis* condensin I complex (Kimura & Hirano, 1997), bind DNA even in the absence of ATP, we incubated condensin with DNA substrates for 20 min in the absence of ATP. As expected, we observed no DNA compaction during this time period (Fig EV2A). We then

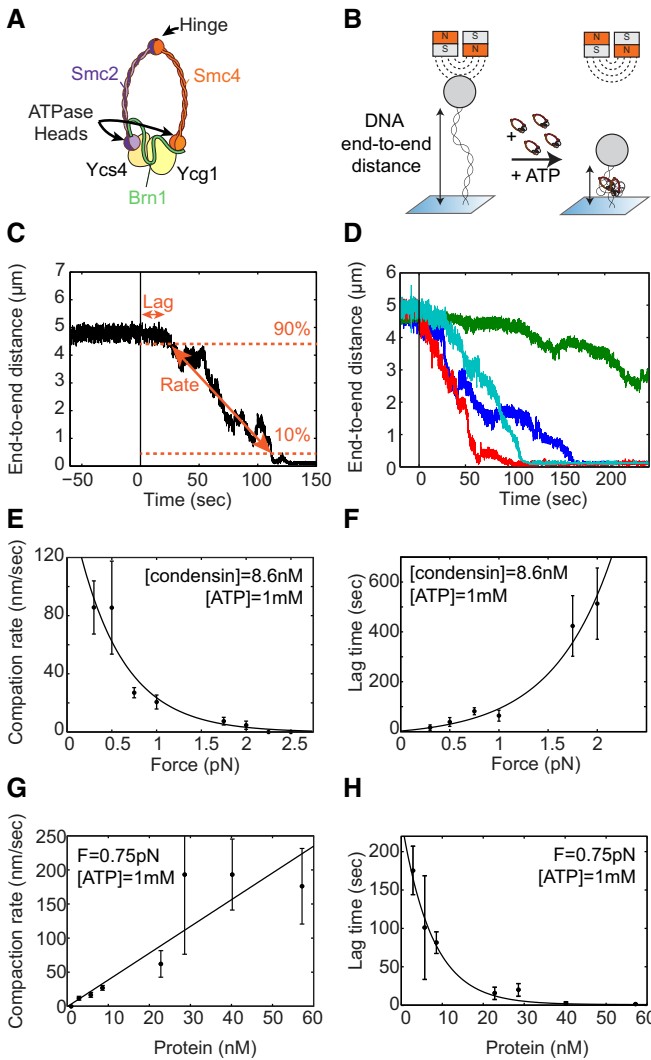

**Figure 1.  Condensin compacts DNA in the presence of ATP.**

A  Cartoon of the yeast condensin complex. Smc2 and Smc4 dimerize via their hinge domains. The kleisin Brn1 associates with the Smc2 and Smc4 head domains to create a ring-like structure. HEAT-repeat subunits Ycs4 and Ycg1 bind to Brn1.

B  Schematic representation of the compaction experiment. A DNA molecule is tethered between a glass slide and a magnetic bead. When condensin and ATP are added, the end-to-end length of the DNA decreases.

C  Characterization of the compaction process with two parameters. The lag time is defined as the time it takes for the compaction to initiate. The compaction rate is set by the compaction speed between 90 and 10% of the original end-to-end length.

D  Examples of compaction traces. Each colour represents a different individual DNA tether measured in the same experiment. Condensin (8.6 nM) and ATP (1 mM) are added at time point zero.

E  The average compaction rate decreases as force increases. At forces higher than 2 pN, condensin does not compact DNA. At 2 pN, two out of nine tethers did not condense. At 1.75 pN, two out of eight tethers did not condense. Error bars represent SEM. For all these experiments, condensin concentration was 8.6 nM and ATP concentration was 1 mM ATP. An exponential curve (line) is added as a guide to the eye.

F  The lag time increases as force increases. Error bars represent SEM. An exponential curve (line) is added as a guide to the eye.

G  The average compaction rate increases linearly as protein concentration increases. Error bars represent SEM.

H  The lag time decreases as protein concentration increases. Error bars represent SEM. An exponential curve (line) is added as guide to the eye.

loading of condensin onto chromosomes (Piazza *et al*, 2014; Kschonsak *et al*, 2017). Indeed, this tetrameric version of condensin showed no DNA compaction activity whatsoever (Fig EV3A). To specifically test the requirement for the Ycg1–Brn1 DNA-binding groove, we repeated the experiment with a version of the condensin holocomplex that contains charge-reversal mutations in the DNA-binding groove. This mutant can still hydrolyse ATP, but its ATPase activity is not stimulated by the presence of DNA (Kschonsak *et al*, 2017). Consistent with the result for the tetrameric complex, this complex was also unable to induce DNA compaction in our assay (Fig EV3B).

To further test whether compaction is due to ATP binding and hydrolysis by the Smc2 and Smc4 subunits of the condensin complex, we purified a version of the condensin complex with point mutations in the Q-loop motifs of the Smc2 and Smc4 ATPase sites ($Smc2_{Q147L}$–$Smc4_{Q302L}$). As expected, the mutant complex was unable to induce DNA compaction in our assay (Figs 2C and EV3C). We then replaced ATP by the only slowly hydrolysable analog ATPγS to distinguish whether the reaction depends on ATP hydrolysis or merely on ATP binding to condensin. Also in this experiment, we observed no DNA compaction (Figs 2C and EV3D), which demonstrates that compaction requires ATP hydrolysis. Finally, we tested whether ATP hydrolysis is required only to initiate compaction or continuously during the active compaction process by exchanging ATP by ATPγS once the DNA had been compacted halfway. In this experiment, compaction did not proceed any further (Fig EV3E). We conclude that both, ATP binding and ATP hydrolysis, are essential for the DNA compaction activity that we observe.

## Condensin remains bound to DNA after force-induced decompaction

We next tested whether the condensin–DNA interaction could be disrupted by applying a high force once the compaction reaction

washed the flow cell with buffer containing no nucleotides to remove all unbound condensin and only then switched to buffer containing 1 mM ATP (but *no* additional protein; hereafter called "sequential addition"). After ATP addition, we observed robust DNA compaction (Fig EV2A, N = 11) with a similar rate as we had measured when we had added protein and ATP simultaneously (Fig 2C). These experiments indicate that condensin binds in the absence of ATP, remains attached during washing steps and can start DNA compaction when ATP is subsequently added (Strick *et al*, 2004). Interestingly, on average lag times were shorter for the sequential addition set-up, although the difference is not statistically significant (Fig 2D). A shorter lag time could suggest that part of the delay that we had observed after protein addition in the standard reaction set-up is due to the time it takes for condensin to bind to the DNA.

To verify that the compaction that we observed was due to the interaction of the condensin complex with DNA in a manner that reflects the physiological properties of condensin function *in vivo*, we tested a tetrameric condensin complex that lacked the Ycg1 HEAT-repeat subunit. Together with the kleisin subunit Brn1, Ycg1 creates a DNA-binding groove in the condensin complex that is essential for the

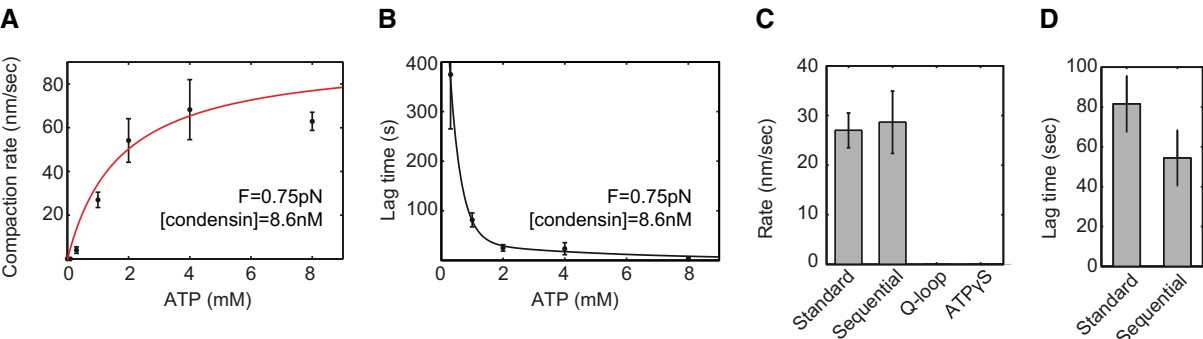

**Figure 2.  Compaction depends on ATP hydrolysis.**

A   Average compaction rate increases according to a Michaelis–Menten relation with ATP concentration. For this protein concentration, the rate saturates at ATP concentrations higher than 2 mM. Error bars are SEM.

B   Lag time decreases as ATP concentration increases. Line is added as guide to the eye. Error bars are SEM.

C   Average compaction rate for the standard experiment (0.75 pN, 8.6 nM protein, 1 mM ATP) and the sequential addition experiment (first 8.6 nM protein/no ATP, wash with buffer, and only then add 1 mM ATP). Rate is similar for standard and sequential addition. The ATPase mutant and wild type in the presence of ATPγS do not show compaction. Error bars are SEM.

D   The lag time for the standard experiment and the sequential addition experiment. The lag time decreases for sequential addition. Error bars are SEM.

had taken place. First, we quantified the end-to-end extension of the bare DNA at 10 and 0.75 pN forces (Fig 3A). After adding condensin and ATP, we observed compaction, as before (Fig 3B). As soon as the DNA molecule had been compacted to about half of its original length, we abruptly increased the force to 10 pN (Fig 3C). Upon this sudden force increase, the end-to-end length did not immediately recover to the fully extended level, in contrast to the response of a bare DNA molecule. Instead, it took a few seconds (~5 s in the example in Fig 3C) until the DNA had extended all the way to the end-to-end length that we had measured for the bare DNA at 10 pN (Fig 3A). When we subsequently lowered the force to 0.75 pN, the DNA started to compact again from the same level it had started at the beginning of the experiment (Fig 3D). We conclude that condensin-dependent DNA compaction can be fully reversed by stretching the DNA with high forces, consistent with a previous report for the *Xenopus* condensin I complex (Strick *et al*, 2004). This, however, does not hinder subsequent compaction against low force.

In the same experiment, we repeated the 10 pN pulling step, and this time it took even longer (~25 s) until the DNA recovered the full end-to-end length (Fig 3E). While keeping the force at 10 pN, we then washed the flow cell with buffer *without* ATP or protein to remove all nucleotide and unbound condensin. When we then lowered the force to 0.75 pN, the DNA did *not* compact (Fig 3F). Strikingly, however, as soon as we added ATP (but *no* additional protein), we again observed compaction (Fig 3G). This result demonstrates that, first, condensin can stay bound to DNA even when stretching the DNA at high forces and washing with physiological buffers and, second, that condensin that had remained bound to DNA requires ATP to initiate a new round of DNA compaction. We confirmed the findings outlined in Fig 3A–G in many independent experiments (*N* = 28).

**Condensin uses two distinct modes to bind DNA**

Condensin might mediate DNA compaction through direct electrostatic interactions with the DNA helix, through topologically

encircling DNA within its ring structure, through pseudo-topologically entrapping DNA by inserting a DNA loop into its ring, or through a combination of these modes (see Discussion). Whereas electrostatic interactions are sensitive to high salt concentrations, topological entrapment of chromosome fibres can resist salt concentrations of 500–1,000 mM NaCl, as shown in bulk biochemistry experiments (Cuylen *et al*, 2011). To assay how condensin interacts with DNA during and after the compaction reaction in our single-molecule set-up, we assayed whether compaction remained stable after washing with buffer containing 500 mM NaCl once condensin had compacted the DNA in an ATP-dependent manner. We found that DNA compaction was fully reversed by the high-salt conditions (Fig 3H, *t* = 450 s, *N* = 7). This indicates that electrostatic interactions with DNA are required for maintaining the condensin-mediated compacted DNA state. Strikingly, when we subsequently lowered salt concentrations to physiological levels (125 mM NaCl) in the presence of ATP (but without adding more protein), we again observed compaction (Fig 3I, *t* = 1,050 s). This demonstrates that condensin, after it had been loaded onto DNA by use of ATP, remained associated with the DNA during the high-salt wash and was capable of again compacting DNA in an ATP-dependent reaction once salt concentrations had been lowered.

We next tested whether ATP was required to allow condensin to bind DNA in a salt-resistant manner. We first incubated condensin with DNA in physiological buffer conditions without ATP (as in the sequential addition experiment). As expected, we observed no compaction in the absence of nucleotide (Fig 3J, *t* = 0–1,300 s). We then washed with high salt buffer (500 mM NaCl) before again lowering salt concentrations to 125 mM and adding ATP (Fig 3K). In contrast to the previous experiment where we had allowed condensin to bind DNA in the presence of ATP before the high-salt wash, we did not observe any compaction (Fig 3L, *N* = 9). Similarly, when we incubated condensin with DNA in the presence of ATPγS instead of ATP before the high-salt wash, we did not observe compaction once we lowered the salt conditions and added ATP (Fig EV2B, *N* = 14). These experiments demonstrate that

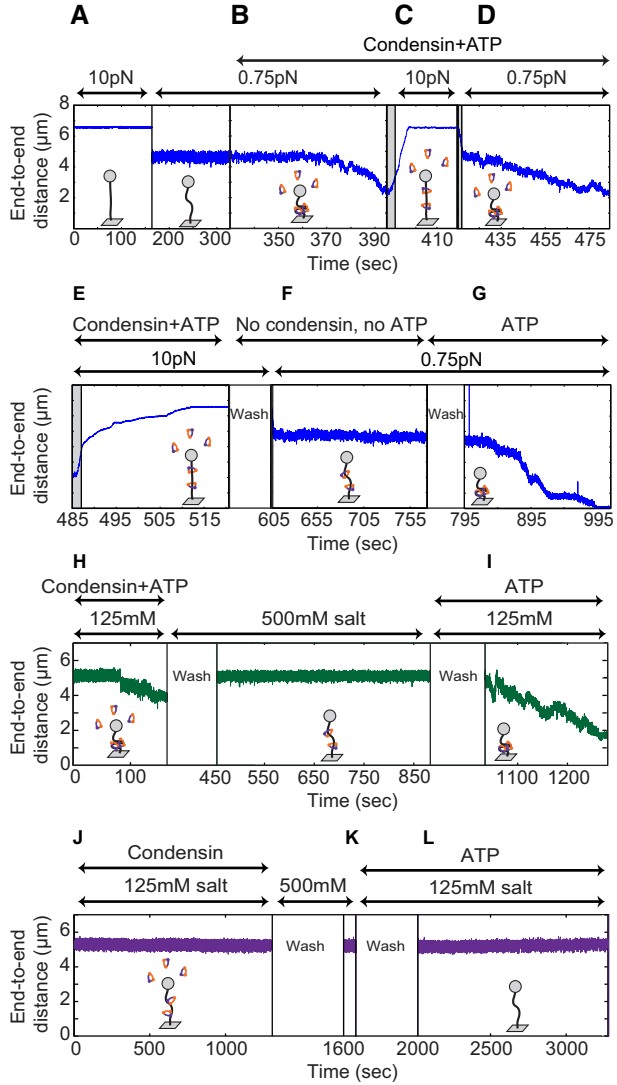

**Figure 3. Decompaction by switching to high force or high salt.**

A Pre-measurements of the DNA tether length before addition of condensin. The average end-to-length was recorded at 10 pN (left) and 0.75 pN (right). This trace is an example taken from *N* = 28 independent experiments. Although different tethers showed different rates of compaction and decompaction, the qualitative result was identical for all.

B With the force at 0.75 pN, condensin (8.6 nM) and ATP (1 mM) were added and the DNA was compacted.

C After about 50% compaction, the force was suddenly increased to 10 pN. The grey area indicates where the time that the magnet was moving, and the black vertical line indicates the point the magnet arrived at the 10 pN position. DNA end-to-end length increased, reversing compaction and eventually recovering the full end-to-end length.

D The force was subsequently lowered to 0.75 pN again. Condensin and ATP were still present and the DNA condensed again.

E The force was increased to 10 pN again. The DNA end-to-end length again increased and eventually recovered to the premeasured full length of bare DNA at 10 pN. Next, the flow cell was washed with buffer without any ATP or protein.

F The force was then lowered to 0.75 pN, and the DNA was observed to not compact in the absence of ATP. Next, the flow cell was washed with buffer with 1 mM ATP but no protein.

G After thus adding ATP but no extra protein, the DNA was able to condense again, indicating that the protein remained bound after pulling and washing.

H The green trace shows a different experiment. At time = 0 s, 8.6 nM condensin and 1 mM ATP were added as normal. After compaction, the flow cell was washed with high salt (500 mM), and the compacted structure was extended again.

I At time = 900 s, the flow cell was washed with physiological salt and 1 mM ATP but no additional protein, and the DNA compacted again.

J The purple trace shows a different experiment. At time = 0 s, 8.6 nM condensin but no ATP was added, and no compaction was observed.

K The flow cell was then washed with high salt (500 mM), and no change in end-to-end length was detected.

L The flow cell was washed with physiological salt and 1 mM ATP was added. No compaction was observed.

that ATP hydrolysis is required to convert condensin from a salt-sensitive to a salt-resistant binding mode, which is indicative of topological binding.

We finally examined whether continued ATP hydrolysis was necessary to maintain DNA in the compacted form, since it had been reported that continuous ATP hydrolysis is necessary to maintain the structure of mitotic chromosomes (Kinoshita *et al*, 2015). When we interrupted ongoing DNA compaction by flushing with buffer without ATP, compaction did neither continue nor reverse. Instead, the DNA end-to-end length remained stable (Fig EV2C, *N* = 9). When we added ATP again, compaction proceeded. These data demonstrate that the presence of ATP is required to initiate and continue compaction, but is neither necessary for maintaining condensin's association with DNA nor essential for preserving already compacted DNA structures.

### Condensin compacts DNA in a stepwise manner

Many compaction traces showed sudden distinct decreases in the DNA end-to-end length, which we will refer to as "steps". We used

a very conservative user-bias-independent step-finding algorithm to extract the size of these compaction steps (see Materials and Methods and Appendix Fig S1 for details). In brief, this algorithm objectively evaluates if a trace displays steps without prior knowledge of step size or location, based on chi-squared minimization. Figure 4A shows a typical example of a DNA compaction trace with fitted steps. We used this hands-off algorithm to analyse all traces we had collected and to determine step sizes.

This analysis revealed that condensin can induce steps of hundreds of nm size (Fig 4B, dark grey). Note that these are remarkably high values, which are clearly larger than the size of the ~50-nm-long condensin molecule itself. The step distribution is very broad, indicating that there is a range of possible outcomes for individual compaction steps. A critical evaluation of the step-size analysis, including cross-checks where we detected simulated steps (for details see Appendix Fig S1), revealed that the distribution analysis is biased towards the observation of larger steps. Small steps in the range of the dimensions of the condensin complex are, in contrast, difficult to detect and are likely to have been missed in the step detection shown in Fig 4B. Indeed, our validation analysis suggests that the real step distribution contains more small steps (Fig 4B, light grey histogram). The fact that we miss these steps in the step-size detection algorithm is mainly due to the noise that is intrinsically large for magnetic tweezers under the low-force conditions required for the compaction experiments. The same limitation holds

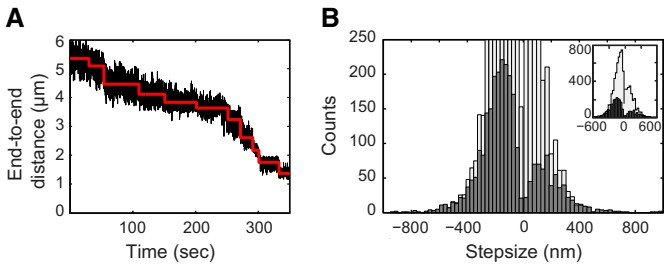

**Figure 4. Condensin compacts DNA in a stepwise manner.**

A  Compaction occurs with discrete steps. Black trace depicts the raw data; red is the fitted step trace.

B  Histogram of detected step sizes (dark grey) and corrected distribution (light grey, see Appendix Fig S1 for details about the correction of the step-size distribution). Inset shows the same histograms displayed to higher counts.

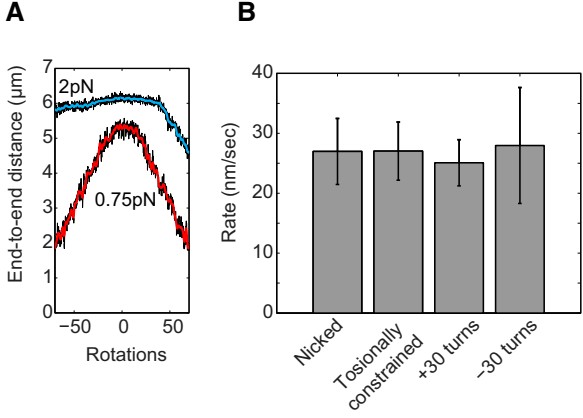

**Figure 5. DNA supercoiling does not influence compaction.**

A  Rotation curves of a bare DNA molecule at constant forces (2 pN in blue, 0.75 pN in red), showing that this molecule is torsionally constrained and supercoils are introduced by applying positive or negative rotations to the magnets.

B  Compaction rates for different supercoiling states. All measurements are for the standard experiment: 0.75 pN, 1 mM ATP, 8.6 nM condensin. There is no difference between nicked DNA ($N = 12$) and coilable DNA ($N = 13$). Also there is no difference between relaxed DNA, and DNA with applied turns in either direction ($N = 8$ for +30, $N = 11$ for −30). Error bars are SEM.

for previously published magnetic tweezer data, although details will depend on data processing, filtering and averaging. Notably, traces resulting from force-induced decompaction (Fig 3, 10 pN) were smooth and did not show any discernible steps (and were accordingly rejected by our step-finding algorithm).

### Condensin does not compact DNA by inducing DNA supercoiling

Since condensin had been reported to influence the supercoiled state of plasmid DNA in the presence of topoisomerases *in vitro* (Kimura & Hirano, 1997; Kimura *et al*, 1999; Bazett-Jones *et al*, 2002; St-Pierre *et al*, 2009), it was proposed that condensin might actively introduce (positive) supercoiling into DNA helices to promote their compaction. We therefore examined the compaction activity as a function of the DNA supercoiling state, an assay for which magnetic tweezers are especially suitable. An example of a rotation curve for a torsionally constrained DNA molecule is shown in Fig 5A. On average, half of the DNA tethers in each experiment showed a decrease in length upon rotation and hence were torsionally constrained, while the other half did not show any decrease in end-to-end length upon rotation due to a nicked tether. When we compared compaction rates between nicked and torsionally constrained DNA molecules, we found no differences (Fig 5B). This finding is fully consistent with the results of an earlier study using *Xenopus* condensin I (Strick *et al*, 2004). In addition, we also tested whether the initial topological state of the DNA affects the compaction process, by introducing +30 or −30 turns into the DNA molecules before adding protein and ATP. Again, we did not find a measurable effect on the compaction rate (Fig 5B).

If the decrease in the end-to-end length during compaction were due to condensin introducing supercoils, we should be able to actually *extend* DNA that had previously been compacted by condensin, as condensin would remove some of the applied supercoils (cf. Seidel *et al*, 2005). We therefore applied +50 or −50 turns to a DNA molecule that was halfway compacted (Fig EV4A). Upon starting the rotation curve in either direction, we never observed that the DNA end-to-end length increased, but instead measured a decrease in compaction in both cases (Fig EV4B). These findings show that the condensin-induced decrease in compaction was not a result of DNA supercoiling.

However, when we rotated the magnet back to the starting position (0 turns) after applying 50 turns to compacted tethers, we found that the end-to-end length did not fully recover. In fact, the end-to-end length started to decrease further already before the "relaxed" point at 0 turns. This behaviour occurred regardless of the direction in which the DNA had initially been rotated ($N = 8$, both directions). We speculate that instead of actively introducing supercoils, condensin is able to "lock" DNA plectonemes by embracing their stem (Fig EV4C).

## Discussion

### DNA binding and compaction are distinct steps in the condensin reaction cycle

We have used magnetic tweezers to characterize the association with and compaction of single DNA molecules by the budding yeast condensin complex. Our results extend previous studies that demonstrated that condensin I complexes immunopurified from *X. laevis* egg extracts are able to compact DNA in magnetic tweezers (Strick *et al*, 2004). In accordance with these and bulk biochemical studies (Kimura & Hirano, 1997; Kimura *et al*, 1999), our data show that association of condensin with DNA can take place in the absence of ATP (Fig EV2A). This ATP-independent interaction is able to survive washing steps with physiological salt concentrations, but it does not survive in buffer conditions of high ionic strength (Fig 3J and K), which indicates that the ATP-independent interaction of condensin with DNA may be electrostatic in nature. Since mutant complexes that prevent DNA binding within the charged Ycg1–Brn1 groove or lack the Ycg1 subunit entirely (Kschonsak *et al*, 2017) show no compaction activity, we propose that this salt-sensitive

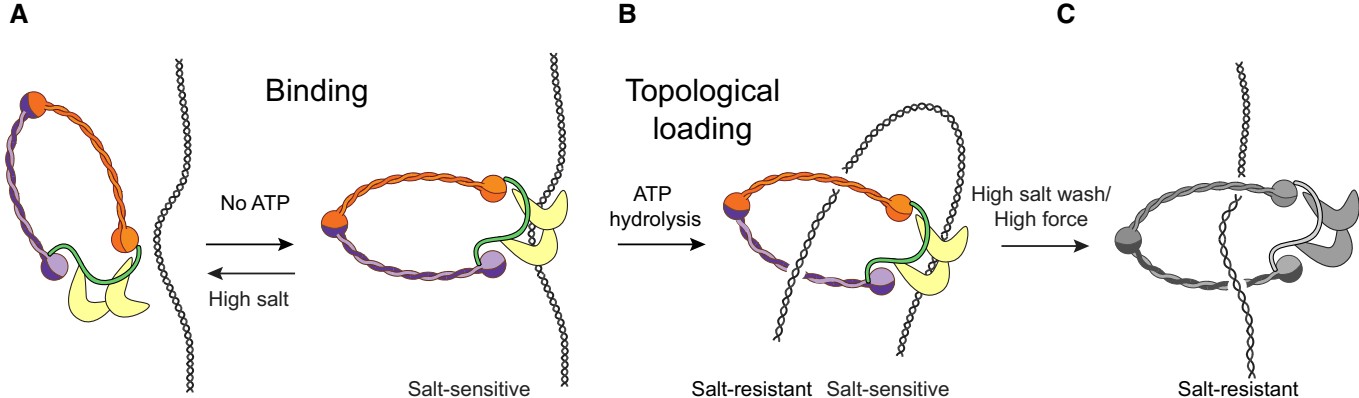

**Figure 6. Condensin compacts DNA using a multistep binding mechanism.**

A   Condensin binds to DNA electrostatically, presumably through the HEAT-repeat subunits.
B   Upon ATP hydrolysis, condensin embraces the DNA topologically, thereby initiating the compaction of DNA.
C   High salt or high force can disrupt the electrostatic interactions in our *in vitro* assays.

binding step occurs through the direct interaction with the DNA double helix of the condensin HEAT-repeat and kleisin subunits (Fig 6A and B). Taking into account that the condensin complexes used in our assays have been purified to homogeneity, these findings suggest that condensin does not require any additional loading factor(s) to associate with and to compact DNA. This contrasts the related cohesin complex, which commonly uses specific loading factors to increase the efficiency of its binding to DNA (Murayama & Uhlmann, 2014, 2017; Davidson *et al*, 2016; Stigler *et al*, 2016).

When condensin is added to DNA in the presence of ATP, it is, however, able to survive high-salt conditions (Fig 3H and I). This suggests that the ATP-dependent mode of DNA binding must be exceptionally stable, for example such as provided by a topological binding mode where the Smc2–Smc4–kleisin condensin ring encircles the DNA. The subsequent compaction step essentially depends on ATP hydrolysis by the Smc2–Smc4 subunits of the condensin complex, since neither a Q-loop ATPase mutant version of condensin in the presence of ATP nor a wild-type version of condensin in the presence of ATPγS are able to compact the tethered DNA substrates in our assay. It thus appears logical to conclude that the electrostatic interaction is converted into a topological interaction by an ATP-dependent temporary ring opening and entry of the bound DNA into the ring (Fig 6C). It is conceivable that the initial electrostatic interaction releases upon ATP hydrolysis, which frees this binding site to be available to grab another piece of the same DNA and thereby create a DNA loop. In summary, our results reveal that at least one electrostatic interaction and a topological interaction must function as the principle binding modes that condensin employs to compact DNA. It is furthermore possible that a third interaction and binding mode is involved in the actual compaction process.

Following a short lag time after addition of ATP, condensin induces a fast compaction of the DNA tethers. We interpret the lag time before compaction starts after addition of condensin and ATP as the sum of the time necessary for condensin to bind to DNA and to become active for compaction. The latter step likely involves the conversion of an electrostatic into a topological binding mode (Fig 6C). This interpretation is consistent with the findings that the lag time depends on the concentration of ATP and is reduced when

condensin has been pre-bound to DNA in the sequential addition set-up. The observation of a lag time is furthermore consistent with recent measurements of condensin movements on DNA curtains, where condensin binds and pauses before becoming active for translocation (Terakawa *et al*, 2017).

Once the compaction reaction had been initiated, it frequently proceeded to the maximally compacted state that we can measure in our set-up. Dominance of DNA compaction over any decompaction in our assays contrasts previous findings with *Xenopus* condensin I complexes, where DNA compaction reverted spontaneously in many instances (Strick *et al*, 2004). Another difference to this previous report is our finding that compaction rates by the *S. cerevisiae* condensin complex scaled approximately linearly with protein concentration. Whereas these data cannot rule out that multiple condensin complexes cooperate in the compaction reaction, they are consistent with a model in which multiple complexes act as individual motors on DNA.

Whereas our experiments strongly suggest that the required energy for compaction must stem from ATP hydrolysis by the Smc2–Smc4 subunits (see above), we find that condensed DNA remains compacted even after washing with buffer that does not contain ATP (Fig EV2C) or buffer that contains ATPγS (Fig EV3E). This shows that continuous ATP hydrolysis by condensin is not required to maintain the compact state of the DNA. The compact state can, however, be disrupted by applying very high physical forces or high-salt conditions, which presumably disrupt non-topological condensin–DNA contacts (Fig 6C).

The amount of work needed for compaction can be calculated as the product of the displacement against the applied force. Taking into account that $k_BT = 4.1$ pN*nm and that the free energy resulting from hydrolysis of one DNA molecule of ATP is ~20$k_BT$, we can calculate the amount of ATP molecules that would minimally be required to drive compaction against a certain force. Assuming for the sake of argument that condensin converts the energy from ATP hydrolysis with 100% efficiency, we estimate that full compaction (from 5 to 0 μm length) against a force of 0.75 pN requires the hydrolysis of 46 ATP molecules, or equivalently, the hydrolysis of each ATP molecule would correspond to a 110-nm step. While this

clearly provides an order-of-magnitude estimate only, the result is consistent with previous estimate of large steps as observed for condensin translocation (Terakawa *et al*, 2017). As the force increases, more ATP needs to be hydrolysed to provide the necessary energy in order to achieve compaction.

A surprising finding from our experiments is the broad distribution of compaction step sizes, which includes very large step sizes, larger than the condensin complex itself. One explanation for this conundrum might be that condensin could be taking smaller individual compaction steps and that the steps that we are detecting are in fact bursts of smaller steps that cannot be resolved within the temporal resolution and noise of the magnetic tweezers assay (0.1–0.4 s, see Appendix Fig S3 and Crut *et al*, 2007). Our step-finding validation demonstrates that, while large steps are confidently detected, the assay is unable to detect very small steps (see Appendix Fig S2). Our conservative method revealed that one should be cautious with step fitting at high compaction activity of condensin and the low forces applied in the magnetic tweezers. For this reason, we refrain from reporting a typical step size for a single condensin-driven compaction cycle.

### Consequences for geometric models for condensin-induced DNA compaction

Which of the various geometric models for condensin's mechanism are compatible with our findings? *Generation of DNA supercoiling* has been proposed as a mechanism to condense DNA (Bazett-Jones *et al*, 2002). Our data are not consistent with this model, since we could never observe unwinding of induced supercoils after compaction (Fig EV4). We also did not find any difference in rates between relaxed DNA, torsionally constrained DNA, and DNA molecules with pre-applied turns (Strick *et al*, 2004). Instead, our results indicate that condensin might stabilize or "lock" plectonemes, for example by binding specifically to crossed DNA segments at the stem of DNA plectonemes (Fig EV4C). However, while such a mechanism would allow condensin to stabilize an already compacted DNA state, it is unable to induce compaction on its own and hence cannot explain the observed compaction activity.

The *random cross-linking* model proposes that condensin compacts DNA by randomly connecting different pieces of the same DNA molecule (Cheng *et al*, 2015) (Fig EV5A). Such a scenario fits well with a broad distribution of step sizes as well as with step sizes that are considerably larger than the dimensions of the condensin complex itself. This model requires, however, that distant DNA regions come into close proximity for cross-linking in the first place, without the action of condensin. Since, at a force of 1 pN, the DNA tethers in our assay are already stretched to 85% of their contour lengths, it is difficult to imagine how, under these forces, large loops could be generated through random cross-linking. Furthermore, since this model does not involve a catalytic compaction activity, it does not explain how halfway compacted DNA molecules can compact further after any free protein has been washed away, as it is quite unlikely that this would happen by condensin letting go of one piece of DNA to grab another piece of DNA further away in order to create a larger loop. Theoretical modelling of the biophysics of a cross-linked DNA polymer under an applied force would be helpful to estimate these notions quantitatively. A variation of the random cross-linking model might involve individual condensin complexes

that mutually interact to generate a DNA loop, that is in a variation of the handcuff-like model that has been proposed for the cohesin complex (Zhang *et al*, 2008). Yet, this model also faces the same challenge of explaining how halfway compacted DNA molecules can continue to compact after any free protein has been washed away.

A model that recently gained much attention is *loop extrusion* (Nasmyth, 2001). Here, condensin binds to DNA and moves it through its ring to extrude a loop of DNA, which thereby continuously increases in size (Fig EV5B). Simulations have shown that loop extrusion can indeed achieve efficient chromosome condensation (Goloborodko *et al*, 2016b). Requirements for this model are that condensin has DNA motor activity, which was demonstrated recently (Terakawa *et al*, 2017), and that the extrusion machine can interact with at least two points along the DNA simultaneously. If the interaction of condensin with DNA would only be topological, loop extrusion would not work, as DNA can slip out of the ring, which certainly will happen under an applied force. Our finding that a direct (electrostatic) contact between condensin and DNA is required to maintain the compacted state of DNA suggests that such a contact might serve as an anchor site at the base of a forming loop (Kschonsak *et al*, 2017). The finding that halfway compacted DNA molecules can eventually compact fully without the addition of extra protein is furthermore easy to imagine for a motor extruding a loop of ever-larger size.

Cartoons of the loop extrusion mechanism often depict a pseudo-topological embrace of the DNA (Fig EV5C). For such pseudo-topological loading, the condensin ring does not necessarily have to open, in contrast to real topological loading. Importantly, we find that a pseudo-topological embrace is inconsistent with our data, as such a conformation would not survive high-salt washes and high force. Instead, our data indicate that the ATP hydrolysis-assisted DNA loading is truly topological. This is an important distinction that changes the way one should think about loop extrusion, and we accordingly suggest that one should take the topologically loaded state as the basis for future modelling of the loop extrusion process (Fig 6).

In conclusion, systematic evaluation of DNA compaction by condensin complexes allowed us to resolve the binding mode conditions that must be met in any geometric model. Our data demonstrate a two-step model: first ATP-independent direct interaction of condensin with DNA, followed by ATP hydrolysis-dependent topological loading and DNA compaction. This model provides an important stride forward in unravelling the mechanism of chromosome compaction by condensin complexes.

## Materials and Methods

### Protein purification

Wild-type (Smc2–Smc4–Brn1–Ycs4–Ycg1), tetrameric (Smc2–Smc4–Brn1–Ycs4), ATPase mutant (Smc2$_{Q147L}$–Smc4$_{Q302L}$–Brn1–Ycs4–Ycg1) and DNA binding mutant (Smc2–Smc4–Brn1$_{K409D, R411D, K414D, K451D, K452D, K456D, K457D}$–Ycs4–Ycg1) versions of the *S. cerevisiae* condensin holocomplex were overexpressed from galactose-inducible promoters in budding yeast. The complexes were purified from interphase cell extracts via a tandem affinity chromatography strategy, using a His$_{12}$ tag fused to the Brn1 subunit and a triple StrepII tag fused to the Smc4 subunit, followed by a gel filtration

step. Expression and purification of the complexes are described in detail in Terakawa *et al* (2017) and Kschonsak *et al* (2017). Fractions from the gel filtration step that corresponded to monomeric condensin holocomplexes were aliquoted, snap-frozen in liquid nitrogen and stored at −80°C.

### Magnetic tweezers

We used a multiplexed magnetic tweezers as described in De Vlaminck *et al* (2012) and Eeftens *et al* (2015). We used a 20-kb DNA construct with digoxygenin- and biotin handles and nitrocellulose-coated flow cells (volume 30 μl) as described in Eeftens *et al* (2015). In brief, nitrocellulose-coated flow cells were incubated with 100 mM anti-digoxygenin antibodies (Fab-fragment, Roche). Then, the flow cell was washed with washing buffer (20 mM Tris–HCl pH 7.4, 5 mM EDTA). Next, the surface was passivated with 10 mg/ml BSA for 1 h and washed again. Streptavidin-coated beads (MyOne, Life Technologies) were incubated with biotin-functionalized DNA for 20 min. After incubation, the beads were washed three times with washing buffer plus 0.05% Tween. An excess amount of beads with digoxygenin-functionalized DNA was then incubated in the flow cell for 10 min. Finally, the flow cell was washed extensively with compaction buffer (10 mM HEPES–NaOH pH 7.9, 125 mM NaCl, 5 mM MgCl$_2$, 1 mM DTT) to flush out all unbound beads and provide near-physiological reaction conditions. Compaction was only observed at conditions around physiological salt concentrations (50–250 mM NaCl, data not shown). Different forces were applied by linear translation of the magnets, while rotation of the magnets was used to apply supercoils. A force calibration curve was generated to correlate the magnet height to the force. Before all experiments, all tethers were routinely checked for coilability and for their end-to-end length before starting the compaction reaction (pre-measurement).

### Determination of the compaction rate and lag time

All compaction experiments were carried out in compaction buffer (10 mM HEPES–NaOH pH 7.9, 125 mM NaCl, 5 mM MgCl$_2$, 1 mM DTT). Different concentrations of ATP and of the *S. cerevisiae* condensin holocomplex (nanomolar range) were dissolved in 50 μl of compaction buffer and flushed in, which typically took 15 s. Tracking of the beads was started immediately after flushing in the protein and the force was kept constant throughout the experiment. The lag time was defined as the time it took for the compaction to start. The time points at which the DNA reached 90, 80, 70%, etc., of their original end-to-end length (taken from the pre-measurement) were automatically recorded by our custom-made software. The compaction rate was determined by calculating the difference in end-to-end length between the 90 and 10% time points. In the case that compaction did not reach the 10% point, we determined the rate from the initial part of the compaction curve. The standard duration of an experiment was 20 min.

### Step analysis

We used a well-defined step-fitting algorithm that was previously described (Kerssemakers *et al*, 2006). This algorithm objectively evaluates if a trace shows steps, without prior knowledge of step

size or location, based on chi-squared minimization. To evaluate the variation of step sizes in an objective manner, we improved the implementation of this algorithm to allow for hands-off, batch style analysis. For details, see Appendix Fig S1.

**Expanded View** for this article is available online.

### Acknowledgements
We thank Ana Mota for preliminary research, Allard Katan, Jekyung Ryu, and Jakub Wiktor for discussions, and Damien D'Amours for plasmids and yeast strains for overexpression of the condensin holocomplex. This work was supported by the ERC Advanced Grant SynDiv (no. 669598 to C.D.), the ERC Consolidator Grant CondStruct (no. 681365 to C.H.H.), and by the Netherlands Organization for Scientific Research (NWO/OCW) as part of the Frontiers of Nanoscience program. S.B. acknowledges support from an EMBL Interdisciplinary Postdoctoral fellowship (EIPOD) under Marie Curie Actions (COFUND).

### Author contributions
JME magnetic tweezer experiments and data analysis. SB and MK protein expression and purification. JME and JK software. JME, SB, CHH and CD conceived and designed the study and wrote the manuscript.

### Conflict of interest
The authors declare that they have no conflict of interest.

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
