## [Review Process File · The EMBO Journal]

Manuscript EMBO-2017-97596

Real-time detection of condensin-driven DNA compaction reveals a multistep binding mechanism

Jorine M. Eeftens, Shveta Bisht, Jacob Kerssemakers, Marc Kschonsak, Christian H. Haering & Cees Dekker

Corresponding author: Christian H. Haering, EMBL & Cees Dekker, Delft University of Technology

Review timeline:

Submission date:	15 June 2017
Editorial Decision:	31 July 2017
Revision received:	26 September 2017
Editorial Decision:	13 October 2017
Revision received:	18 October 2017
Accepted:	19 October 2017

Editor: Hartmut Vodermaier

Transaction Report:

1st Editorial Decision

31 July 2017

Thank you for submitting your manuscript on single molecule and mechanistic analyses of purified condensin to The EMBO Journal. We have now received comments from three expert referees, copied below for your information, as well as feedback from an expert member of our Editorial Advisory Board. As you will see, the referees generally appreciate your careful and dedicated analyses and their technical quality, but referees 1 and 2 (as well as our expert advisor) also raise the well-taken concern that partially similar conclusions have to some extent already been reached by an earlier study from Strick et al (2004). In this light, we are unfortunately not convinced that the manuscript offers a sufficiently major advance to warrant EMBO Journal publication, at least in its current form.

That said, we realize that the study may well become a more compelling candidate for an EMBO Journal article if extended/deepened through the various straightforward experimental suggestions made in particular by referee 2. Should you be prepared to undertake such further experiments (as well as to address the various more specific technical and presentational issues raised), then we should be happy to consider a revised manuscript further for publication. I should point out that it is our policy that competing manuscripts that might be published elsewhere during revision at EMBO Press will have no negative impact on our final assessment of a revised study, thus giving you an opportunity to diligently follow-up on all the various points raised by the referees.

Should you choose to submit an extended study to The EMBO Journal, please note that the editorial process at the stage of resubmission would be greatly facilitated by proper formatting of the revised manuscript according to EMBO Journal author guidelines, in particular regarding referencing, expanded view and/or appendix figures and materials, as well as by suggestions for 2-5 one-

sentence 'bullet points' with brief factual statements that summarize key aspects of the paper, which would form the basis of an online 'Synopsis' for the article. Additional information on preparing and uploading a revision can also be found below.

Please do not hesitate to get in touch with me in case you should have any questions/comments regarding the referee reports or the revision requirements; we might also arrange for an extended revision period if this should be helpful.

Thank you again for the opportunity to consider this work for The EMBO Journal. I look forward to hearing from you.

Referee reports

Referee #1:

The manuscript from Eeftens et al., employs magnetic tweezers to measure real time compaction of DNA using condensin purified from yeast *S. cerevisiae*. Several clear findings emerge using this elegant technique. Compaction via condensin proceeds in large (200 nm) steps, the compaction rate is dependent upon the ATP hydrolysis and condensin makes an electrostatic interaction with DNA that is ATP independent (although ATP does stabilise this interaction). Based on the findings the authors propose a two step process for condensin binding DNA through electrostatic interactions firstly and then ATP hydrolysis to entrap and then drive compaction. The science in general is solid and the writing clear. My main issue with this manuscript is several key findings (effect of ATP binding, hydrolysis, supercoiled DNA) have already been demonstrated by the Hirano lab (Strick T, Current Biology, 2004) also using nanomanipulation but with purified condensin from *Xenopus* mitotic egg extracts. Although I am all for verifying results independently, it is important to discuss previous overlapping work in detail to enable the reader to ascertain what is new, what is being verified or added to. The experiments showing the variation of salt/ ATP binding and the effect on condensin association/compaction are interesting and I like the model but it is not entirely clear to me whether there is sufficient advance over previous studies. I have detailed these concerns and others below.

Materials and Methods: Little detail of the condensin purification is given. I would have expected an image of the purified intact condensin complex to be presented given it forms the basis for all experiments. Furthermore it wasn't stated whether the condensin was purified from M phase or not. I would assume the mitotic condensin would be most relevant to the study? As such, I was surprised condensin purified from interphase and mitotic phases were not compared throughout.

(Smc2-Q147L/Smc4-Q302L): Have these mutants been described previously or characterised, i.e., have they been shown unable to hydrolyse ATP or is this being inferred based on amino acid conservation? The effect in cells would also be of interest: are the mutations lethal, does condensin carrying the mutations load onto chromosomes *in vivo*?

Discussion: The author's state: "In contrast to this previous report (Strick T, Current Biology, 2004), DNA compaction clearly dominated any de-compaction in our assays" and then "Our data show that association of condensin with DNA can take place in the absence of ATP". This later statement (perhaps not intentionally) reads as if it is a novel result when several *in vitro* studies including the stated (Strick T, Current Biology, 2004, Kimura K, Cell, 1999) show condensin ATP binding is not necessary for DNA binding. These should be referenced appropriately. It would be also useful here to discuss why (in contrast) two independent studies (Kinoshita et al, Dev Cell, 2015; Hudson et al, MBC, 2008) show SMC2/4 subunits bearing ATP binding mutations are unable to associate with assembled chromosomes.

Aside from the similar findings for ATP binding, the Strick paper also examined ATP hydrolysis and found (similarly) that ATP hydrolysis is essential for compaction using the non-hydrolysable analogue AMP-PNP (instead of ATP γ S used in current manuscript). Likewise the effect of supercoiled DNA was analysed in both studies and similar results were seen. Strick paper: "we observe similar behavior whether the DNA is torsionally relaxed, positively supercoiled, or

negatively supercoiled. Similar results were also obtained on nicked DNA"; Eeftens paper: "Remarkably, we found no differences in the compaction rates between nicked and torsionally constrained DNA molecules". Again this reads as a novel result when there appears to be significant overlap with the earlier study.

Referee #2:

The manuscript by the Dekker and Haering laboratories investigates the mechanism of DNA compaction by the yeast condensin complex *in vitro* under defined conditions using magnetic tweezers. The presented work is related to a study by Strick and Hirano (2004) which demonstrated ATP hydrolysis dependent DNA compaction by a *Xenopus* condensin complex using a similar experimental system. Here, the authors provide additional data for example by exploring in more detail the force-dependence of DNA compaction and its salt-resistance.

The manuscript is presented in a very professional manner. The data is of high quality and easy to access. The work addresses an important question in a dynamic research field. However, considering the already available literature, the impact of the findings may be somewhat limited. Furthermore, some of the main conclusions of the work are not sufficiently supported by the presented data (yet) (see below).

Main concerns:

Condensin has two modes of interaction with DNA (electrostatic and topological) (Piazza et al., 2014; Cuylen et al., 2011). The authors claim that both binding modes are essential for DNA compaction *in vitro* (as for example stated in the last sentence of the abstract). However, while an electrostatic interaction is clearly present, its requirement for the DNA compaction process is not established by this report. The data show that an electrostatic interaction is needed to keep condensin bound to DNA if low-salt washes are performed prior to the addition of ATP. Whether this interaction is important during DNA compaction itself (in the presence of ATP) is left unclear (because high-salt washes disrupt the electrostatic interaction and hinder the ATP-dependent DNA compaction).

The Haering laboratory has recently mapped an interface for DNA at one of condensin's heat repeat subunits (Piazza et al., 2014). To address the issue described above, mutants that block this DNA interaction site (Piazza et al., 2014) should be tested in magnetic tweezer experiments. These experiments will show whether salt-sensitive DNA binding is indeed dependent on this interface and whether DNA compaction relies on DNA binding at this interface. This should be an easy experiment for the Dekker/Haering labs.

The rate of condensation increases with protein concentration, which the authors interpret as multiple condensin complexes individually contributing to the condensation of a single DNA molecule. Are condensation steps more frequent at higher protein concentration? Do longer DNA molecules feature faster compaction, while shorter ones condensin DNA more slowly. These are simple predictions that should be tested. Why do condensin complexes make smaller steps when more highly concentrated?

The step size distribution is presented for several experimental conditions in Figure 4 and S4. However, only the low force and low salt condition (Figure 4) includes positive steps (that is DNA decompaction). Why is this? Is DNA decompaction due to slippage of DNA through condensin? If so, then the size of positive steps should be highly force dependent (even at the 0-2 pN range)? Please add this crucial information to the respective figure panels.

Q-loop mutants are rarely used in the study of condensin and other SMC protein complexes. A short explanation and a basic characterization of these mutants (or citations of relevant data) is needed.

The authors show that the presence of ATP is dispensable for the maintenance of previously established DNA compaction. More importantly and more interestingly, is continued ATP hydrolysis needed for progressive DNA compaction or is it only needed during the initiation of DNA compaction (for example during loading)? The former would be expected if condensin acts as

a (DNA extrusion) motor. The authors should be able to address this question by substituting ATP for ATPgS during on-going DNA compaction.

The authors argue that their data is inconsistent with pseudo-topological loading of condensin onto DNA, because such a conformation would not survive high salt washes and high force (page 10, 3rd paragraph). These strong claims need to be supported by data or references. DNA is relatively stiff and may remain trapped pseudo-topologically even at high force (when DNA loops become even more unfavorable) and high salt (which may further stiffen DNA). Moreover, none of the models shown in Figure 6 and Figure S6 explain how DNA compaction by condensin might force-resistant as at least one DNA fiber is always freely diffusing in the SMC ring.

Some of the citations in the manuscript refer to preprint manuscripts that have not yet been peer-reviewed. This should be somehow made clear in the text. Especially, in cases where strong conclusions are made from the preprints (page 3, top paragraph; page 8, 3rd paragraph; page 10 middle paragraph).

The effect of the topological state on DNA compaction has already been studied (with similar conclusions) in Strick et al., 2004. A reference is thus needed in the relevant paragraph on page 7 (Figure 5).

Referee #3:

The preprint entitled Real-time detection of condensin-driven DNA compaction reveals a multistep binding mechanism describes a series of well design single molecule experiments addressing the possible operating mode of condensing.

The paper is very well written and the different assays presented are demonstrative and bring valuable insights on the condensin action mode that could not be obtained in bulk assay. The methodology is simple and convincing and the results not straight forward. Thus the work definitely shed new light on the subject and is worth publishing.

I have no serious critics to raise, the logic of the experiments is convincing and the work very serious. The point which I find the weakest is the step size measurement, the data treatment is always tricky on that subject and the observation of a step size greater than the condensin one is certainly a concern for me. The authors discuss of a possible bunching of steps of different condensin, but the argument is somewhat weak. If some improvements are required on the paper certainly that point could be addressed.

The absence of supercoiling implication in the condensing mechanism is convincing and certainly new, the assay is simple but demonstrative.

Referee #1:

The manuscript from Eeftens et al., employs magnetic tweezers to measure real time compaction of DNA using condensin purified from yeast *S. cerevisiae*. Several clear findings emerge using this elegant technique. Compaction via condensin proceeds in large (200 nm) steps, the compaction rate is dependent upon the ATP hydrolysis and condensin makes an electrostatic interaction with DNA that is ATP independent (although ATP does stabilise this interaction). Based on the findings the authors propose a two step process for condensin binding DNA through electrostatic interactions firstly and then ATP hydrolysis to entrap and then drive compaction. The science in general is solid and the writing clear.

We thank the Reviewer for emphasizing the clear findings of our work.

My main issue with this manuscript is several key findings (effect of ATP binding, hydrolysis, supercoiled DNA) have already been demonstrated by the Hirano lab (Strick T, Current Biology, 2004) also using nanomanipulation but with purified condensin from *Xenopus* mitotic egg extracts. Although I am all for verifying results independently, it is important to discuss previous overlapping work in detail to enable the reader to ascertain what is new, what is being verified or added to. The experiments showing the variation of salt/ ATP binding and the effect on condensin association/compaction are interesting and I like the model but it is not entirely clear to me whether there is sufficient advance over previous studies. I have detailed these concerns and others below.

We agree with the Reviewer that our work builds upon the earlier work by Strick *et al.* 2004. Obviously, we do know this work very well, and while we greatly appreciate this pioneering paper, we strongly feel that our rigorous testing of many parameters provides key new insights that go well beyond these early results. In particular, in our paper, for the first time, ...

... we provide a quantitative analysis of kinetic parameters for the compaction reaction, its dependence on ATP and protein concentrations, and describe how compaction depends on force. The fact that compaction is completely abolished at forces above a very modest force of 2pN is unexpected and remarkable for a molecular DNA motor and will have to be taken into account for the mechanism of chromosome condensation.

... we demonstrate that the compaction activity is due to (a) condensin binding to DNA and (b) the ATPase activity of the condensin protein complex, as we show by including specific mutant complexes (including newly added data in the revised manuscript). Validation by specifically engineered mutations has not been possible previously.

... we show that condensin remains bound to DNA during (reversible) decompaction by applying high force or high salt concentrations and that it is subsequently able to recompact DNA when the force or salt concentrations are lowered. This has essential implications for the molecular model of how condensin compacts DNA.

... we resolve the controversy on possible supercoiling induced by condensin. With careful measurements, we show that condensin does not induce supercoils into DNA but likely rather acts by “locking” supercoils.

Taken together, our findings demonstrate a physical picture of a multistep binding model for condensin-driven compaction, which we feel is of urgent interest to the chromosome biology field. Given the confusion in the field, we believe that our experiments that clearly lay out such a multistep loading into a true topologically bound state (as opposed to a generally depicted pseudo-topologically bound state) are of great importance.

In the revised manuscript, we have now referenced the Stick *et al.* paper more extensively and better clarified which aspects of our work are confirmatory and which aspects are new compared to this previous work.

Materials and Methods: Little detail of the condensin purification is given. I would have expected an image of the purified intact condensin complex to be presented given it forms the basis for all experiments.

We are providing detailed information about the condensin complex purification and the biochemical characterization of wild-type and mutant complexes used in our work in two recent publications: Terekawa *et al.*, Science 2017 eaan6516 (earlier provided as a pre-print on bioRxiv) and Kschonsak *et al.*, Cell 2017 in press (attached as a copy to this manuscript). To avoid duplication of these data, we are referring the reader to these papers.

Furthermore it wasn't stated whether the condensin was purified from M phase or not. I would assume the mitotic condensin would be most relevant to the study? As such, I was surprised condensin purified from interphase and mitotic phases were not compared throughout.

Our protein complexes are overexpressed in budding yeast cells and then purified via several chromatography steps. At the time of harvest, most cells are in G1 phase of the cell cycle and the purified proteins will hence be most likely without mitotic modifications. With this system, it is not technically feasible to purify large amounts of condensin protein complexes from mitotically arrested cells. We have added a short note explaining that proteins were purified from interphase extracts to the Methods section.

Note that this approach enables us, for the first time, to also test mutant complexes, which we believe is a valuable addition to our manuscript.

(Smc2-Q147L/Smc4-Q302L): Have these mutants been described previously or characterised, i.e., have they been shown unable to hydrolyse ATP or is this being inferred based on amino acid conservation? The effect in cells would also be of interest: are the mutations lethal, does condensin carrying the mutations load onto chromosomes *in vivo*?

The Smc2_{Q147L}–Smc4_{Q302L} mutant has mutations in the γ -phosphate switch loops (Q-loops) in both Smc2 and Smc4. This mutant is still able to bind DNA but has no ATP hydrolysis activity, as we demonstrate in Terekawa *et al.*, Science 2017 ean6516 (earlier provided as a pre-print on bioRxiv).

Discussion: The author's state: "In contrast to this previous report (Strick T, Current Biology, 2004), DNA compaction clearly dominated any de-compaction in our assays" and then "Our data show that association of condensin with DNA can take place in the absence of ATP". This later statement (perhaps not intentionally) reads as if it is a novel result when several *in vitro* studies including the stated (Strick T, Current Biology, 2004, Kimura K, Cell, 1999) show condensin ATP binding is not necessary for DNA binding. These should be referenced appropriately.

The Reviewer is correct on the latter point: ATP-independent binding of condensin to DNA was shown by Strick *et al.* and others before. We apologize for perhaps giving the wrong impression. We now clarified this point in the revised manuscript and added the appropriate reference (page 8, second paragraph).

Note, however, that we maintain our point that, in contrast to the experiments described by Strick *et al.*, our data show dominant compaction over de-compaction.

It would be also useful here to discuss why (in contrast) two independent studies (Kinoshita *et al.*, Dev Cell, 2015; Hudson *et al.*, MBC, 2008) show SMC2/4 subunits bearing ATP binding mutations are unable to associate with assembled chromosomes.

Kinoshita *et al.* and Hudson *et al.* reported that Walker A mutant complexes were unable to associate with chromatin in extracts or in cultured cells, respectively. We noticed during the purification of an *S. cerevisiae* Walker A condensin complex that this mutant complex is prone to aggregation during size exclusion chromatography, most likely because mutation of surface-exposed lysine residues in the P-loops to hydrophobic isoleucine residues (used in both of these publications) results in a (partial) misfolding of the SMC head domains. This explains why these proteins did not associate with chromosomes in the previous experiments. For this reason, we have decided to use the Q-loop mutant complex in our study, which, in contrast to the Walker A mutant complex, shows the same biochemical characteristics as the wild-type complex.

Since the read-out of our assay is the DNA end-to-end distance, our assay does not allow the direct observation whether condensin associates with the DNA. Instead, we observe condensin's effect on the compaction of DNA. For this reason, we do not make a statement whether or not our mutant complexes are able to bind DNA.

Aside from the similar findings for ATP binding, the Strick paper also examined ATP hydrolysis and found (similarly) that ATP hydrolysis is essential for compaction using the non-hydrolysable analogue AMP-PNP (instead of ATP γ S used in current manuscript). Likewise the effect of supercoiled DNA was analysed in both studies and similar results were seen. Strick paper: "we observe similar behavior whether the DNA is torsionally relaxed, positively supercoiled, or negatively supercoiled. Similar results were

also obtained on nicked DNA "; Eeftens paper: "Remarkably, we found no differences in the compaction rates between nicked and torsionally constrained DNA molecules". Again this reads as a novel result when there appears to be significant overlap with the earlier study.

We apologize for evoking the incorrect impression of claiming novelty at this specific point. We corrected this in the revised manuscript. We note, however, that Strick *et al.* referred to the effect of the supercoiling state of DNA on the measured step size distribution, whereas we are measuring compaction rates. Whilst the two conclusions are related, they are not identical.

Referee #2:

The manuscript by the Dekker and Haering laboratories investigates the mechanism of DNA compaction by the yeast condensin complex in vitro under defined conditions using magnetic tweezers. The presented work is related to a study by Strick and Hirano (2004) which demonstrated ATP hydrolysis dependent DNA compaction by a *Xenopus* condensin complex using a similar experimental system. Here, the authors provide additional data for example by exploring in more detail the force-dependence of DNA compaction and its salt-resistance.

The manuscript is presented in a very professional manner. The data is of high quality and easy to access. The work addresses an important question in a dynamic research field. However, considering the already available literature, the impact of the findings may be somewhat limited. Furthermore, some of the main conclusions of the work are not sufficiently supported by the presented data (yet) (see below).

We thank the Reviewer for acknowledging the importance of the subject and our contributions.

Regarding the point on how our work expands previous work we feel that our manuscript provides novel insights that go significantly beyond the pioneering results of Strick et al. While we greatly appreciate this paper, we strongly feel that our rigorous testing of many parameters provides key new insights that go well beyond these early results. In particular, in our paper, for the first time, ...

... we provide a quantitative analysis of kinetic parameters for the compaction reaction, its dependence on ATP and protein concentrations, and describe how compaction depends on force. The fact that compaction is completely abolished at forces above a very modest force of 2 pN is unexpected and remarkable for a molecular DNA motor and will have to be taken into account for the mechanism of chromosome condensation.

... we demonstrate that the compaction activity is due to (a) condensin binding to DNA and (b) the ATPase activity of the condensin protein complex, as we show by including specific mutant complexes (including newly added data in the revised manuscript). Validation by specifically engineered mutations has not been possible previously.

... we show that condensin remains bound to DNA during (reversible) decompaction by applying high force or high salt concentrations and that it is subsequently able to recompact DNA when the force or salt concentrations are lowered. This has essential implications for the molecular model of how condensin compacts DNA.

... we resolve the controversy on possible supercoiling induced by condensin. With careful measurements, we show that condensin does not induce supercoils into DNA but likely rather acts by “locking” supercoils.

Taken together, our findings demonstrate a physical picture of a multistep binding model for condensin-driven compaction, which we feel is of urgent interest to the chromosome biology field. Given the confusion in the field, we believe that our experiments that clearly lay out such a multistep loading into a true topologically bound state (as opposed to a generally depicted pseudo-topologically bound state) are of great importance.

Main concerns:

Condensin has two modes of interaction with DNA (electrostatic and topological) (Piazza et al., 2014; Cuylen et al., 2011). The authors claim that both binding modes are essential for DNA compaction *in vitro* (as for example stated in the last sentence of the abstract). However, while an electrostatic interaction is clearly present, its requirement for the DNA compaction process is not established by this report. The data show that an electrostatic interaction is needed to keep condensin bound to DNA if low-salt washes are performed prior to the addition of ATP. Whether this interaction is important during DNA compaction itself (in the presence of ATP) is left unclear (because high-salt washes disrupt the electrostatic interaction and hinder the ATP-dependent DNA compaction).

Indeed, as the Reviewer states, our salt wash experiments demonstrate (a) that the electrostatic interaction is needed for condensin's interaction with DNA (Fig. 3J) and (b) that the high salt wash interrupts ongoing compaction (Fig. 3H-I). This shows that DNA molecules that are already compacted still need this electrostatic interaction to stay compacted.

The Haering laboratory has recently mapped an interface for DNA at one of condensin's heat repeat subunits (Piazza et al., 2014). To address the issue described above, mutants that block this DNA interaction site (Piazza et al., 2014) should be tested in magnetic tweezer experiments. These experiments will show whether salt-sensitive DNA binding is indeed dependent on this interface and whether DNA compaction relies on DNA binding at this interface. This should be an easy experiment for the Dekker/Haering labs.

We thank the Reviewer for this valuable suggestion. We have now tested two additional mutants: a tetrameric complex that lacks the DNA interacting subunit Ycg1 and a mutant with charge-reversal mutations in the newly identified DNA binding groove formed by the Brn1 and Ycg1 subunits, described in Kschonsak *et al.*, Cell 2017 in press (attached as a copy to this manuscript). Both, the DNA-binding mutant and the tetramer, show no compaction. We have added the findings on both mutants to the manuscript text and show the new data as Figure EV3A and B.

The rate of condensation increases with protein concentration, which the authors interpret as multiple condensin complexes individually contributing to the condensation of a single DNA molecule. Are condensation steps more frequent at higher protein concentration? Do longer DNA molecules feature faster compaction, while shorter ones condensin DNA more slowly. These are simple predictions that should be tested. Why do condensin complexes make smaller steps when more highly concentrated?

Indeed, the rate of compaction increases with higher protein concentration and condensation steps are more frequent. In the revised manuscript, we have significantly expanded the validation for our step finding results (see also response to Reviewer #3), and we mention now more explicitly that we cannot accurately assess the precise step size. To avoid over-interpretation and prevent speculations that are not sufficiently supported by the data, we have decided to adopt a conservative approach and removed the figure where we showed differences in apparent step size for the different conditions.

Due to technical limitations, we are limited in the length range of molecules that we can measure on. We cannot make torsionally constrained tweezer-constructs longer than 20kb with sufficient yield. We did test shorter DNA molecules (8kb instead of 20kb) and find that the compaction rate is not significantly different ($p=0.27$). We chose to measure compaction on the longest DNA possible in order to observe the most compaction features per experiment. We include measurements of the relative rate of compaction while tethers are being shortened (Figure EV1), and show that the end-to-end distance has no influence on the compaction rate that we report.

The step size distribution is presented for several experimental conditions in Figure 4 and S4. However, only the low force and low salt condition (Figure 4) includes positive steps (that is DNA decompaction). Why is this? Is DNA decompaction due to slippage of DNA through condensin? If so, then the size of positive steps should be highly force dependent (even at the 0-2 pN range)? Please add this crucial information to the respective figure panels.

This appears to be a misunderstanding: we find positive steps at all conditions tested, not only at low force and salt. We now clarify in the manuscript that we find positive steps in all conditions. Generally, positive steps are a lot less frequent than the negative steps (Fig 4B). For fast compaction conditions (high protein concentration, low force), we find these positive steps less frequently. Unfortunately, the datasets for the positive steps per condition are numerically too small to perform a statistically significant analysis. The positive steps indeed signal de-compaction, but it highly depends on the underlying model if this should be imagined as slippage through the ring, release of a loop, or otherwise. We prefer to refrain from speculations about the mechanistic origin of the positive steps. Please also note that we have expanded the step-analysis and have adjusted our claims (see previous point).

Q-loop mutants are rarely used in the study of condensin and other SMC protein complexes. A short explanation and a basic characterization of these mutants (or citations of relevant data) is needed.

The Smc2_{Q147L}-Smc4_{Q302L} mutant complex has mutations in the γ -phosphate switch loops (Q-loops) in both Smc2 and Smc4. This mutant is still able to bind DNA but has no ATP hydrolysis activity, as we demonstrate in Terekawa *et al.*, Science 2017 ean6516 (earlier provided as a pre-print on bioRxiv). We prefer using the Q-loop mutant over the more frequently used Walker A mutation due to potential misfolding issues of the latter (for

details, see response to Reviewer #1).

The authors show that the presence of ATP is dispensable for the maintenance of previously established DNA compaction. More importantly and more interestingly, is continued ATP hydrolysis needed for progressive DNA compaction or is it only needed during the initiation of DNA compaction (for example during loading)? The former would be expected if condensin acts as a (DNA extrusion) motor. The authors should be able to address this question by substituting ATP for ATP γ S during on-going DNA compaction.

The Reviewer correctly notes that, interestingly, ATP is dispensable for the maintenance of previously established DNA compaction. We thank him/her for the suggested experiment of substituting ATP γ S during on-going DNA compaction. We have now carried out this experiment and include the results in the new Figure EV3E. We see that compaction does *not* continue after we substitute ATP with ATP γ S, indicating that ATP hydrolysis is necessary for continuation of compaction after condensin loading. We now discuss this finding in the main text on page 4.

The authors argue that their data is inconsistent with pseudo-topological loading of condensin onto DNA, because such a conformation would not survive high salt washes and high force (page 10, 3rd paragraph). These strong claims need to be supported by data or references. DNA is relatively stiff and may remain trapped pseudo-topologically even at high force (when DNA loops become even more unfavorable) and high salt (which may further stiffen DNA). Moreover, none of the models shown in Figure 6 and Figure S6 explain how DNA compaction by condensin might force-resistant as at least one DNA fiber is always freely diffusing in the SMC ring.

We stand by our conclusion that a pseudo-topological loading is not consistent with our data for a number of reasons:

1. In Figure 3A-H, we show that high force can completely reverse compaction and recover the full original end-to-end distance of the DNA. This directly shows that DNA does not remain trapped, because DNA would not recover its original end-to-end distance if such features would remain. Indeed, DNA would slip out of a pseudo-topologically bound molecule at these forces.
2. Very different experiments also indicate that DNA can slip in and out tiny rings in a folded fashion. For example, in the nanopore field, it is very well established that DNA will enter a, say, 10nm-wide nanopore in a folded fashion for modest driving forces (\sim 10-20pN electrophoretic force in that case, see e.g. Storm *et al.*, Phys Rev E 71, 051903, 2005). This indicates that the referee's intuition that – due its stiffness – DNA may remain trapped pseudo-topologically at high force is not correct.
3. The persistence length of DNA does vary in a minor way only (from 51 to 55nm) with ionic strength in the range that we are measuring (Baumann *et al.*, PNAS, 1997).

In Figure 6, we mean to visualize the principal binding modes that condensin employs to compact DNA, based on our findings: at least one electrostatic interaction, and a topological interaction. The figure is not intended to claim that the entrapped DNA is unbound or free to slide. We have clarified in the revised manuscript that there most

likely exist additional direct DNA binding sites in the complex that must interact with DNA during the compaction cycle.

Some of the citations in the manuscript refer to preprint manuscripts that have not yet been peer-reviewed. This should be somehow made clear in the text. Especially, in cases where strong conclusions are made from the preprints (page 3, top paragraph; page 8, 3rd paragraph; page 10 middle paragraph).

The bioRxiv manuscript that we referred to has now been peer reviewed and published in Science (Terekawa *et al.*, Science 2017 ean6516). We are furthermore attaching a manuscript that is in press in Cell and describes the new mutant complexes that we have used in the revised manuscript (Kschonsak *et al.*, Cell 2017 in press).

The effect of the topological state on DNA compaction has already been studied (with similar conclusions) in Strick *et al.*, 2004. A reference is thus needed in the relevant paragraph on page 7 (Figure 5).

We agree and added the reference in the relevant section, see page 7.

Referee #3:

The preprint entitled Real-time detection of condensin-driven DNA compaction reveals a multistep binding mechanism describes a series of well design single molecule experiments addressing the possible operating mode of condensin.

The paper is very well written and the different assays presented are demonstrative and bring valuable insights on the condensin action mode that could not be obtained in bulk assay. The methodology is simple and convincing and the results not straight forward. Thus the work definitely shed new light on the subject and is worth publishing.

We thank the Reviewer for the very positive assessment and the recommendation to publish the work.

I have no serious critics to raise, the logic of the experiments is convincing and the work very serious. The point which I find the weakest is the step size measurement, the data treatment is always tricky on that subject and the observation of a step size greater than the condensin one is certainly a concern for me. The authors discuss of a possible bunching of steps of different condensin, but the argument is somewhat weak. If some improvements are required on the paper certainly that point could be addressed.

We thoroughly are aware of the strengths and weaknesses of step size analyses – which is one reason that we involved Jacob Kerssemakers as a co-author, as he likely is the most-cited world expert in step-size analysis (cf. his classic paper Nature 442, 709, 2006). We have examined potential steps in the data traces using a routine that removes user bias to the maximum extent (involving an analysis that perhaps is more thorough than any other single-molecule study so far). The results have led us to believe, after many crosschecks, that the condensation traces indeed exhibit steps.

Prompted by the comment of the referee, we have now expanded the step size analysis even further and added additional validation data in the revised manuscript and appendix. Briefly, we added simulated steps onto experimental base-line traces, and then ran our step-finding algorithm over those traces. We find that, due to technical limitations associated with the noise that intrinsically is relatively large under the experimental conditions that we (and others) use in magnetic tweezers, we are unable to resolve all steps, in particular steps of smaller sizes. Large steps are mostly still well resolved, maintaining the point we make in the paper. The additional analysis allows to present the virtues and limitations of the step size analysis much more explicitly and clearly.

Accordingly, we have now expanded the discussion and the appendix. We also note that it is difficult to compare our results to previous studies side-by-side, because data processing, filtering, and averaging can strongly influence the outcome.

Finally, we note that we present a thorough analysis of the steps, but we do not in any way emphasize the importance of steps, cf. the abstract of our paper. Indeed, the main message of the paper lies in the multistep binding model for condensin-driven compaction.

The absence of supercoiling implication in the condensing mechanism is convincing and certainly new, the assay is simple but demonstrative.

We thank the Reviewer for acknowledging the novel aspect of these findings.

Thank you again for submitting your revised manuscript to our journal. Referee 2 has now assessed it once more, and generally considers the manuscript substantially improved. S/he however retains a number of concerns that would still need to be addressed (esp. the first two points) or at least further discussed (third point) before publication. I am therefore returning the study to you once more for a final round of minor revision, in order to allow you to take care of these issues.

REFEREE REPORT

Referee #2:

The authors have responded to the comments by the reviewers. They made several modifications to the text and figures, which have improved the manuscript. Somewhat surprisingly, they have toned down their interpretation of the step size distribution.

There are some points that need further attention:

Figure 2C and EV3D: Does ATPgS bind to yeast condensin with reasonable affinity and promote head association? If not, the following statement on page 5 is invalid: "Also in this experiment, we observed no DNA compaction (Fig 2C and EV3D), which demonstrates that compaction requires ATP hydrolysis." Please explain and add reference or data.

The authors have added data showing that a DNA binding defective mutant of condensin fails to compact DNA (Fig. EV3). Since the result of the experiment is negative, it is important to mention that the mutant is still an active ATPase (Konshak et al.). The nature of the mutations is hidden in materials and methods (please add to the figure or figure legends). Currently, it is very tedious to link this result to the related experiments shown in Konshak et al.

The authors stand by their conclusion that a pseudo-topological loading is inconsistent with their data. However, their arguments remain soft in the opinion of this referee. The authors make assumptions about the size/circumference of the DNA entrapping structure in condensin, although we have little knowledge about it. While their results may slightly favour a topological type of association, if wrong, these claims can be quite distracting and detrimental. At the same time, they offer little benefits to the field (even if true), since the idea of topological entrapment already dominates the field for several years. I do not see the point of making such strong claims (for example in the abstract).

Referee #2:

The authors have responded to the comments by the reviewers. They made several modifications to the text and figures, which have improved the manuscript. Somewhat surprisingly, they have toned down their interpretation of the step size distribution.

We thank the Reviewer for emphasizing that our manuscript has improved.

There are some points that need further attention:

Figure 2C and EV3D: Does ATPγS bind to yeast condensin with reasonable affinity and promote head association? If not, the following statement on page 5 is invalid: "Also in this experiment, we observed no DNA compaction (Fig 2C and EV3D), which demonstrates that compaction requires ATP hydrolysis." Please explain and add reference or data.

Since ABC ATPases, including the SMC head domains, don't possess a pronounced ATP-binding pocket but instead sandwich a pair of nucleotides between relatively flat surfaces, nucleotide binding affinities are inherently difficult to measure. However, the facts that (a) the replacement of an oxygen in ATP with a sulfur atom in ATPγS will only cause minimal structural differences, (b) the main interactions are made with the adenine base sitting in the P-loop of the protein, and (c) ATPγS binds and promotes head engagement of cohesin Smc3 head domains (Gligoris *et al.*, Science 2014), make it very likely that the Smc2–Smc4 heads can bind and dimerize via ATPγS. We are therefore confident that we can conclude from our data that ATP-hydrolysis is necessary for DNA compaction.

The authors have added data showing that a DNA binding defective mutant of condensin fails to compact DNA (Fig. EV3). Since the result of the experiment is negative, it is important to mention that the mutant is still an active ATPase (Konshak *et al.*). The nature of the mutations is hidden in materials and methods (please add to the figure or figure legends). Currently, it is very tedious to link this result to the related experiments shown in Konshak *et al.*

We already explained the nature of the mutation in the text ("a version of the condensin holocomplex that contains charge-reversal mutations in the DNA-binding groove") and now also added a sentence in the manuscript explaining that the DNA-binding mutant is indeed still active for basal ATP hydrolysis, but that this activity cannot be stimulated by DNA in this mutant.

The authors stand by their conclusion that a pseudo-topological loading is inconsistent with their data. However, their arguments remain soft in the opinion of this referee. The authors make assumptions about the size/circumference of the DNA entrapping structure in condensin, although we have little knowledge about it. While their results may slightly favour a topological type of association, if wrong, these claims can be quite distracting and detrimental. At the same time, they offer little benefits to the field (even if true), since the idea of topological entrapment already dominates the field for several years. I do not see the point of making such strong claims (for example in the abstract).

On this point, we respectfully disagree with the reviewer. We stand by our conclusion based on the arguments in the previous round of reviews: our data is inconsistent with a pseudo-topological loading mechanism.

Note that our arguments for this do not in any way depend on the circumference of the DNA-entrapping ring. Instead, the main point indicating this is that, in Figure 3A-H, we show that high force can completely reverse compaction and recover the full original end-to-end distance of the DNA. This directly shows that DNA does not remain trapped, because DNA would not recover its original end-to-end distance if such features would remain. Indeed, DNA would slip out of a pseudo-topologically bound molecule at these forces. In addition, we see that compaction can occur after this high force pulling and subsequently washing the flowcell to remove unbound protein. These findings are not compatible with a pseudo-topological binding mode.

Furthermore, we believe that our claims do provide a benefit to the field, as nowadays, unlike what the reviewer states, SMC proteins are often depicted in a pseudo-topological

embrace (Hirano, Cell 2016, Dekker&Mirny, Cell 2016, Dolgin, Nature 2017) . By contrast, our data indicate true topological loading, so we feel that it is important to show this to the community.

Thank you for submitting your final revised manuscript for our consideration. I am pleased to inform you that we have now accepted it for publication in The EMBO Journal.

Corresponding Author Name: Christian Haering&Cees Dekker

Journal Submitted to: The EMBO journal

Manuscript Number: EMBOJ-2017-97596